# Upstream watershed condition predicts rural children's health across 35 developing countries

Diego Herrera[1,2,9], Alicia Ellis[3], Brendan Fisher[1,2], Christopher D. Golden[4], Kiersten Johnson[5], Mark Mulligan[6], Alexander Pfaff[7], Timothy Treuer[8] & Taylor H. Ricketts [1,2]

Diarrheal disease (DD) due to contaminated water is a major cause of child mortality globally. Forests and wetlands can provide ecosystem services that help maintain water quality. To understand the connections between land cover and childhood DD, we compiled a database of 293,362 children in 35 countries with information on health, socioeconomic factors, climate, and watershed condition. Using hierarchical models, here we find that higher upstream tree cover is associated with lower probability of DD downstream. This effect is significant for rural households but not for urban households, suggesting differing dependence on watershed conditions. In rural areas, the effect of a 30% increase in upstream tree cover is similar to the effect of improved sanitation, but smaller than the effect of improved water source, wealth or education. We conclude that maintaining natural capital within watersheds can be an important public health investment, especially for populations with low levels of built capital.

[1] Gund Institute for Environment, University of Vermont, 617 Main Street Burlington, Burlington, VT 05405, USA. [2] Rubenstein School of Environment and Natural Resources, University of Vermont, Aiken Center 81 Carrigan Drive Burlington, Burlington, VT 05405, USA. [3] Duke Clinical Research Institute, Duke University, 2400 Pratt St Durham, Durham, NC 27705, USA. [4] Department of Environmental Health, Harvard T.H. Chan School of Public Health, 677 Huntington Ave, Boston, MA 02115, USA. [5] USAID Bureau for Food Security, 1300 Pennsylvania Ave NW, Washington, DC 20004, USA. [6] Department of Geography, King's College London, London WC2R 2LS, UK. [7] Sanford School of Public Policy, Duke University, 201 Science Dr, Durham, NC 27708, USA. [8] Department of Ecology and Evolutionary Biology, Princeton University, 117 Eno Hall Princeton, Princeton, NJ 08544, USA. [9] Present address: Environmental Defense Fund, 1875 Connecticut Ave NW # 600, Washington, DC 20009, USA. Correspondence and requests for materials should be addressed to D.H. (email: dherrera@edf.org)

Research on ecosystem services has highlighted the various benefits that humans derive from nature[1–3]. However, the relationship between ecosystems and perhaps the most fundamental aspect of well-being, human health, is less well understood[4]. The recent Rockefeller Foundation-Lancet Commission on Planetary Health argues that the health impacts from natural systems transformation are still poorly characterized, that the disease burden associated with such alterations is growing, and that improvements in human welfare achieved in the past are likely to be reversed if the current trends of environmental degradation continue[5].

Today more than one in four deaths of children under 5 years of age are attributable to unhealthy environments[6]. Diarrheal disease (DD), the second leading cause of death among children in this age group, is responsible for 361,000 children deaths every year as a result of poor access to clean water, sanitation, and hygiene[7, 8]. Evidence from in situ studies has linked contaminated surface water to DD[9]. This points to watershed degradation as a global environmental and development concern.

Watershed degradation, defined here as the loss of natural land cover and the resulting impacts on hydrology[10], is linked to the loss of essential ecosystem services. Ecosystems such as forests and wetlands can filter pollutants and pathogens from surface water supplies[11, 12]. These systems also contribute fewer nutrients and other pollutants to streams, compared with most human land uses[13, 14], and they can stabilize soil, minimizing erosion, and sediment loading[15, 16]. Therefore, conversion of forest to agriculture or housing can increase pollution, which in turn can have negative effects on the water quality downstream[17, 18].

Significant proportions of the population in the developing world continue to use rivers, lakes, ponds, and irrigation canals as their main source of drinking water[19, 20]. Moreover, there are major urban–rural disparities in access to improved water. As of 2015, 79% of people in urban areas globally have piped water sources, compared only 33% in rural areas[20]. Fully 93% of people using surface water live in rural areas. This suggests that the potential human health costs attributable to watershed degradation may not be distributed equally across urban and rural regions. Livelihoods in rural areas rely more directly on local resources and thus are more dependent on the natural conditions[4, 21]. Urban dwellers, in contrast, have more built infrastructure and complex supply systems drawing a mix of surface and ground water from multiple locations, both close and far from the city[22].

Establishing a relationship between environmental change and human health outcomes is a challenge because health is influenced by many factors—demographic, socioeconomic, environmental, infrastructure, and governance—that often interact with each other. Moreover, behavioral responses can "insulate" humans from impacts, at least when a community has access to capital and education[4]. For example, households that boil water can mitigate the effects of degraded water sources. Despite these analytical challenges, it is essential to determine if improved ecosystems management could function as a legitimate public health investment.

Sanitation infrastructure, improved drinking water sources, education and income are already well known to significantly reduce DD[23–28]. More recently, studies have begun to provide evidence on linkages between the natural environment and human health, including multiple health outcomes and potential confounding factors[29]. However, few have looked at environment–health relationships with multi-country data (but see refs. [30], [31]), and even fewer have evaluated differential impacts across varied contexts, e.g., urban vs rural. Because most of the evidence remains limited to specific countries or case studies, it is difficult to generalize results to other contexts or the globe.

Here we address these knowledge gaps by studying the relationship between upstream watershed conditions and the probability of DD among children under age 5, using a large, geo-referenced data set for 35 developing countries and 293,362 individual children (Fig. 1). Our data set includes individual and household-level health and socioeconomic information from the Demographic and Health Surveys (DHS) program[32]. Key socioeconomic control variables included in our analyses are age of the child, education of the mother, wealth of the household, and access to improved sanitation and water sources (see Methods for details).

Our main environmental variables of interest are two measures of watershed conditions. The first variable reflects the hydrological influence of upstream livestock and people on water quality downstream, hereafter: human activity. The second reflects the hydrological influence of upstream tree cover on water quality downstream, hereafter: tree cover. These variables are based on spatial hydrologic networks, water balance, and land cover to estimate the percentage of the water reaching each cluster of households that was influenced by upstream people and livestock (as potential sources of pollution) or tree cover (see Methods). We also control for climate factors that have been shown to affect the probability of DD, i.e., temperature and precipitation 1 month lags[33] and changes.

We ask three core questions. First, controlling for socioeconomic determinants of health, what effect do upstream tree cover and human activity have on the probability of DD among children downstream? Second, how does this relationship vary between urban vs rural households? Third, how does the

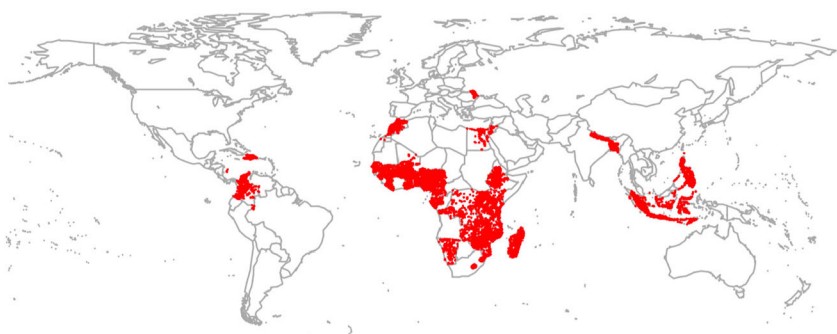

**Fig. 1** Location of household clusters in the sample. Our health and environmental database covers 293,362 individuals in 35 countries (highlighted in red). The database links 2001–2012 geo-referenced Demographic and Health Survey's (DHS) individual and household information with data on temperature, precipitation, and upstream watershed conditions

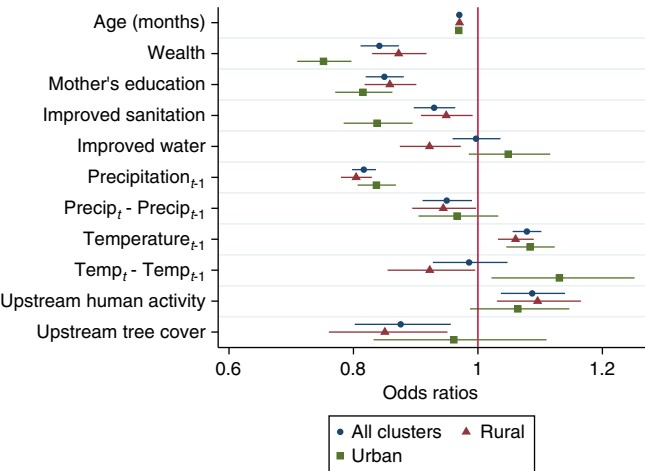

**Fig. 2** Factors associated with the probability childhood diarrheal disease. Variables reducing the probability of diarrheal disease (DD) have odds ratios lower than 1 to the left of the red vertical line. Odds ratios and 95% confidence intervals (horizontal lines) show statistically significant associations between socioeconomic variables and the probability of DD in urban and rural households. The bottom two variables indicate that the associations between upstream watershed variables (human activity and tree cover) and the probability of DD are significant (confidence intervals do not cross the red vertical line) for rural, but not urban households. See Supplementary Table 1 for model coefficients

impact of upstream tree cover compare to that of other factors and potential policy options? We find that upstream tree cover is associated with lower probability of childhood DD in rural areas, an effect that may be comparable in magnitude to that of key socioeconomic factors. Quantifying the role natural conditions play in reducing the probability of DD can help illuminate whether and where conservation efforts may constitute investments in public health.

## Results

**Analysis of urban and rural households**. First we evaluate the sign and the statistical significance of the variables in the model on the probability of DD. Using the full data set, we find significant effects of several explanatory variables on the probability of DD (Fig. 2; Supplementary Table 1). In particular, upstream tree cover is associated with lower probability of DD, consistent with previous literature on forest impacts[34], and supporting the argument that upstream forests can play an important role in terms of regulating water quality. Conversely, upstream human activity is associated with an increase in DD, consistent with earlier findings that human and livestock presence upstream is related to increased contamination of water and thus more DD downstream[35].

In our sample of 35 countries we find significant differences in socioeconomic variables between urban and rural areas (Supplementary Table 2). For example, while 63% of urban households have improved water, only 22% of rural households do. Therefore, we expect that the results of the model and the effect of watershed conditions could differ between these subsets. Fitting the same model to these two groups we find that for rural households, the effects of both upstream human activity and tree cover are significant, while for urban households these effects are nonsignificant (Fig. 2; Supplementary Table 1). For rural households we find significant negative associations between DD and all socioeconomic variables. For urban households the magnitude of the effects of socioeconomic variables appears to be even larger except for improved water which is statistically

nonsignificant. Higher precipitation (1 month lag) is associated with a lower probability of DD, while higher temperature (1 month lag) is associated with a higher probability of DD.

**Comparing key policy options**. To compare the size of the effect of different policy options, we use the model coefficients to calculate marginal effects of changes in wealth, education, improved sanitation, improved water, and tree cover (Fig. 3). For the socioeconomic variables, marginal effects represent the change in the probability of DD from discrete changes, i.e., a wealth level in the two highest quintiles compared with lower quintiles, a level of education of high school or higher compared with not having completed high school, and improved sanitation and water compared with unimproved (see details in Methods). For the continuous tree cover variable, we use the marginal effect to approximate the change in the probability of DD given increases of 10, 20, and 30% in the hydrological influence of upstream tree cover.

In rural areas, upstream tree cover is significant for all three levels of increase (Fig. 3). Education shows the largest impact (a reduction of 13% from a baseline probability of DD of 10.72%), followed by wealth (12% reduction), improved water (7% reduction), and improved sanitation (4% reduction). The effect of a 30% increase in upstream tree cover (4% reduction in the probability of DD) is similar to the effect of an improved sanitation facility, but lower than the effect of improved water, education, and wealth. In urban areas, improved water and upstream tree cover do not have a significant effect in urban households (Fig. 3). Wealth has the largest impact (a reduction of 25% from a baseline probability of DD of 9.74%) followed by education (18% reduction) and improved sanitation (16% reduction).

**Interactions between green and gray infrastructure**. To further explore the interactions between watershed variables and built infrastructure in rural areas, we split the sample of rural households into those with improved and unimproved water. Consistent with the urban vs rural comparison, the effects of upstream tree cover and human activity are nonsignificant in areas with improved water, but significant in households with unimproved water (Fig. 4; Supplementary Table 1). In particular, upstream tree cover reduces the probability of DD in rural households with unimproved water, which suggests that tree cover could be more beneficial for populations with less access to built capital such as piped municipal water supplies.

**Robustness checks**. We run additional robustness checks to explore the interactions between tree cover and human activity upstream in rural areas. We split our rural sample into subgroups of high and low (above and below average) levels of upstream human activity, and re-fit our model for each subgroup (Fig. 5; Supplementary Table 3). Our main result (i.e., a negative association between upstream tree cover and the probability of DD) holds for both areas of high and low upstream human activity, suggesting multiple mechanisms for the effect of tree cover.

Within this same analysis, we explore an alternative model specification including age as binary categorical variables and upstream tree cover as binary variables representing quartiles. The age categorical variables show a nonlinear relationship between the age of the child and the probability of DD, as previously observed in the literature[36]. Relative to the first year of age, children in the second year of age have higher probability of DD, while those older than 2 years of age have lower probability OF DD. Upstream tree cover quartile variables show that children from rural areas with levels of tree cover above the 75th percentile have odds of childhood DD about 20% lower relative to those

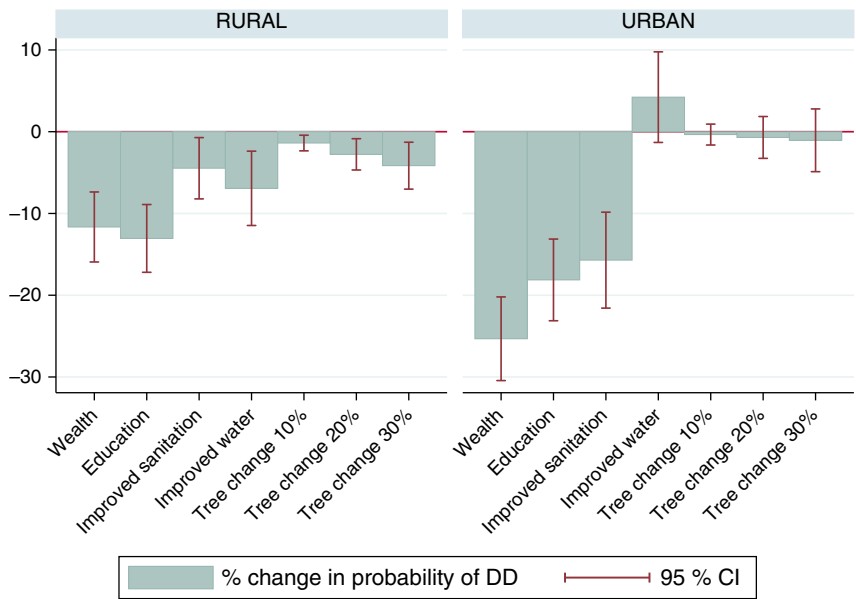

**Fig. 3** Marginal effects of policy options for reducing childhood diarrheal disease. Reductions in the probability of diarrhea and 95% confidence intervals for wealth, education, improved sanitation, improved water, and increases in tree cover of 10, 20, and 30%. Marginal effects of tree cover are only statistically significant for the rural subset. The effect of a 30% increase in tree cover is approximately the same as the effect of an improved sanitation facility, but lower than the effect of improved water, education, and wealth

with tree cover below the 25th percentile (Fig. 5; Supplementary Table 3).

Finally, we test our model for a subset of 10 countries in West and Central Africa: Burkina Faso, Ghana, Guinea, Liberia, Mali, Nigeria, Senegal, Cameroon, Democratic Republic of the Congo, and Gabon. Here we also split the sample into urban and rural households (Supplementary Fig. 1). As we discuss below, these countries are of particular interest for our study given high rates of childhood DD, as well as considerable threats to forests and water resources. The model for this subset shows similar results as our full sample analysis. Both tree cover and human activity are significant in rural households but not in urban households.

## Discussion

Ecosystems and the services they provide are under threat due to high rates of degradation worldwide[37, 38]. Awareness of our dependence upon well-functioning ecosystems is rapidly growing, as is the scientific evidence for it and its use in policy arguments[39, 40]. In this study we provide evidence that watershed condition is associated with measurable health outcomes downstream. In particular, forests can have positive effects on human well-being through reducing childhood DD, especially in rural areas with low levels of built capital. Our study responds to recent reviews[5, 41] by clarifying linkages between ecosystem changes and health outcomes, while accounting for the complex nature of the relationship. This information can help inform strategies to jointly address multiple development objectives, including the new Sustainable Development Goals[42].

We find statistically significant effects of watershed conditions on childhood DD downstream. Controlling for other relevant factors, upstream human activities are associated with increases in DD, while upstream tree cover is associated with reduced DD (Fig. 2). These results are consistent with existing evidence on the health impacts of improved sources of drinking water and sanitation[23–25, 43], as well as income and education[27, 44].

Upstream tree cover could influence the probability of DD downstream through two main mechanisms: (i) by displacing human activities that can pollute the watershed, or (ii) by filtering or diluting pollutants from areas of human activity[11]. Other mechanisms may also contribute; for example, forested watersheds likely have larger water bodies, which also dilute contaminants. Disentangling these mechanisms is difficult with an observational study, but there are two lines of evidence suggesting that trees are doing more than simply displacing people and livestock. First, we find no correlation between upstream tree cover and human activity in our data set (nonsignificant correlation coefficient of −0.0036). Second, the negative relationship between tree cover and DD holds for households with both high and low upstream human activity, suggesting that the effect of tree cover applies even with significant development in the watershed. Determining specific mechanisms will require more detailed studies within specific watersheds, including samples of water quality associated with different mixes of land use.

The patterns observed in our temperature and precipitation control variables correspond with those found in other studies[33, 37, 43, 45]. In particular, higher temperature is linked to increases in DD, while precipitation associated with decreases. Precipitation results are consistent with a dilution effect, whereby high rainfall keeps concentrations of pollutants low. Upstream human activity is likely associated with childhood DD via contamination of water falling on and passing through human populations and associated livestock.

Our comparison of urban and rural regions shows that the relationship between tree cover and DD downstream can vary across different socioeconomic contexts—as should be the case when there are multiple options for assuring health outcomes but those options vary across settings. On average, tree cover is statistically significant only in rural regions, where populations tend to be more dependent on natural conditions (Fig. 2). Conversely, urban areas are likely to be less reliant on local surface water and therefore watershed condition, probably because urban dwellers are not consuming water directly from upstream sources. However, there could be other benefits four urban areas not analyzed in this study in the form of reduced water treatment costs[46, 47].

Comparing potential policy options, we find that in rural areas, the effect of a 30% increase in upstream tree cover is significant

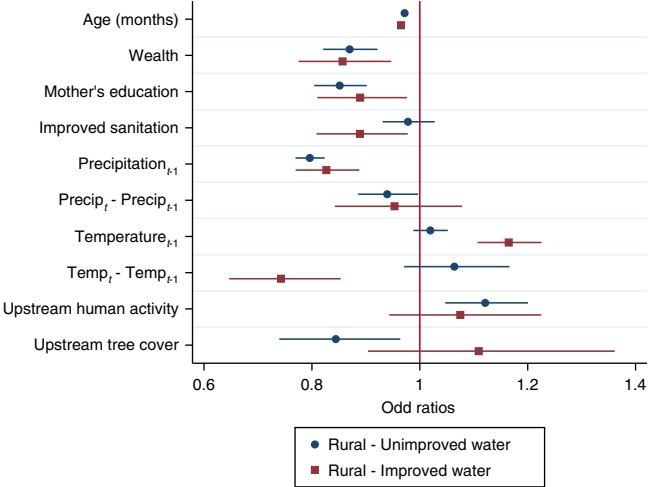

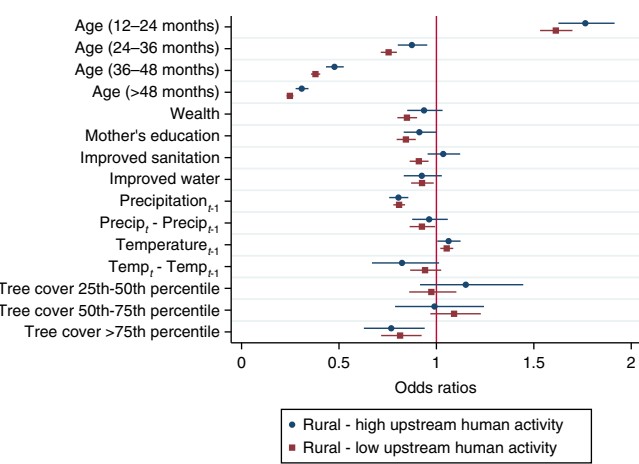

**Fig. 4** Factors associated with childhood diarrheal disease differ between rural households with improved and unimproved water. Variables reducing the probability of diarrheal disease (DD) have odds ratios lower than 1 to the left of the red vertical line. Odds ratios and 95% confidence intervals (horizontal lines) show statistically significant associations (confidence intervals do not cross the red vertical line) between socioeconomic variables and the probability of DD in both groups. However, the associations between watershed variables (human activity and tree cover) and the probability of DD are significant for rural households with unimproved water, but not for rural households with improved water. See Supplementary Table 1 for model coefficients

**Fig. 5** Factors associated with diarrheal disease in rural areas with high and low upstream human activity, including age and upstream tree cover dummy variables. Variables reducing the probability of DD have odds ratios lower than 1 to the left of the red vertical line. Odds ratios and 95% confidence intervals (horizontal lines) show a statistically significant (confidence intervals do not cross the vertical line) negative association between upstream tree cover and the probability of DD for both areas of high and low upstream human activity, suggesting multiple mechanisms for the effect of tree cover. See Supplementary Table 3 for model coefficients

and is similar to the effect of an improved sanitation facility. That said, larger increases in tree cover would be necessary in rural areas to match the effects of high education, high wealth or improved water on the probability of DD (Fig. 3). Within rural regions, tree cover has a significant effect in households without access to improved water sources (Fig. 4). These results together highlight the potential for natural ecosystems to benefit health outcomes in the absence of insulating factors, without understating the importance of human and built capital.

We also find similar effect of upstream tree cover (and urban–rural differences) in a subset of 10 West and Central African countries (Supplementary Fig 1). The African continent, in particular the sub-Saharan region, faces major challenges with potable groundwater quality and threats to the sustained use of aquifers[48]. However, this region has been found to have the potential to provide high ecosystem service values to a population with high rates of poverty[49]. West and Central Africa hold two of the major river basins in the continent (Niger and Congo basins) and account for 95% of African rainforests[50]. These regions have experienced high deforestation rates in the past, which could continue to be a major issue due to population growth, urbanization, and road building[51]. With respect to health, DD is a major cause of mortality among young children in sub-Saharan Africa[52] and countries in the West and Central regions like Nigeria and Congo Democratic Republic top the list of nations contributing to global childhood deaths caused by DD[53]. For all these reasons we consider West and Central Africa as priority regions to test our approach. In these regions, watershed and forest protection strategies could complement efforts to build water and sanitation infrastructure as suggested by previous studies[54].

Several limitations to our study deserve mention here. Our analysis is limited by the set of observable factors available, as well as the countries that have been included in recent DHS surveys. Moreover, although we control for some of the major

determinants of DD and make use of detailed socioeconomic and environmental data, our study is an observational one and should be interpreted as such. We analyze broad trends in the determinants of DD, but similar empirical strategies could be used for finer-scale evaluations (e.g., analysis of priority basins) where specific policy decisions are made. With more detailed information on hydrology, forests, land use change, vegetation, policy interventions, and climate, the effects of tree cover could be disaggregated at the local level to test alternative mechanisms linking tree cover and health outcomes, to understand the independent effects of forests and hydrologic factors, and to model different scenarios of ecosystem change. Our temperature and precipitation variables control for climate conditions of survey month based on long-term monthly averages, but this assumption only holds if the 2000–2012 period does not depart from long-term climate in a given region.

Finally, it is important to note that while human activities such as cattle production are sources of pollution that threaten downstream health, these activities are also sources of income and nutrition that improve the well-being of the populations who have access to them. We are not accounting for such effects in this study. To fully determine if conservation is an effective means of protecting public health, a full cost-benefit analysis comparing different interventions (improving water, sanitation, etc.) is required. This is an important avenue of future research.

Despite these limitations, our analysis highlights the critical role that ecosystem services can play in directly supporting human health and welfare. We provide evidence across many developing countries that conservation strategies, under certain environmental and socioeconomic conditions, could serve as public health investments.

## Methods

**Health and environment database.** The DHS Program, sponsored by USAID, provides technical assistance for the implementation of nationally representative, stratified, two-stage cluster sample household surveys that collect data on population, health, and nutrition for over 90 developing countries around the world[32]. DHS data for this analysis were downloaded from the DHS website in September of 2015. We extracted and processed surveys that took place between 2001 and 2012 with available geospatial coordinates of cluster locations in order

to link the DHS data to external data sets on climate and the environment (Supplementary Tables 4 and 5). Notably, the DHS program does not report exact coordinates for the clusters included in the survey, but randomly displaces the coordinates up to 2 km for urban clusters, and up to 5 km for rural clusters, with a further 1% of rural clusters displaced up to 10 km. This is done to protect the anonymity of the individuals in the survey. For each country and survey, we obtained DHS variables hypothesized to be important risk factors for the probability of DD in children and for which there exists empirical evidence. In this study we analyze data for 35 countries with all variables of interest. The final data set includes 293,362 children that are residents of selected households and are alive at the time of interview.

For each cluster of households in the DHS data, we extracted several climate and environmental variables that can influence the probability of DD and merged them to the DHS data using each cluster's geographic coordinates. We generated the environmental data at a 10 km grid resolution to approximate the environmental conditions for each cluster and address the possible displacement of the exact locations. The environmental variables were averaged within that 10 km area. We used data on monthly average temperature and precipitation as climate controls. Both temperature and precipitation have been shown to significantly affect DD with a 1-month lag[33], so this is the specification that we used in our model. We also included changes in these climate variables between the survey month and the previous month in our model.

*Diarrheal disease*: the DHS instrument specifically asks whether each child had DD in the last 24 h or within the last 2 weeks. We use this information to construct a binary outcome variable that is equal to 1 if the child had DD within the last 2 weeks and 0 otherwise.

*Age*: the DHS surveys provide the age of each child in months, allowing us to control for its effect on DD. All of the children in our data set are under 5 years of age with country averages between 2 and 3 years.

*Wealth and education*: wealth and education are expected to have a significant negative relationship with DD[26, 44]. They can influence access to food, nutrition, health services, medicines, and hygiene practices. The DHS data includes a wealth index, which is a composite measure of a household's living standard that places households into categories representing wealth quintiles. The surveys also provide the level of education of the mother. This is a standardized variable providing level of education in the following categories: No education, Primary, Secondary, and Higher. With this information we defined two binary variables, one for households with a level of wealth in the two highest quintiles, and another for mothers of the children in the study sample with secondary education or higher.

*Improved sanitation and water*: improved sanitation infrastructure is a key factor reducing the probability of DD[25, 55]. DHS identifies the main source of drinking water used by the household, and the type of sanitary facility primarily used by each household. We group the type of water and sanitation used by the household into dichotomous measures reflecting improved or unimproved sanitation and water source based on WHO/JMP definitions, which allow comparisons across countries with more confidence[56]. Supplementary Table 6 shows the categories that we used as improved or unimproved for both water and sanitation.

*Precipitation and temperature*: these two environmental variables have shown to be significant predictors of DD[44, 55]. The temperature variable in our data set is the long-term (1950–2000) mean temperature in the cluster during the survey month in degrees Celsius. Precipitation is measured in millimeters and represents the long-term (1950–2000) mean precipitation in each cluster during the survey month. Both variables were included in the model as standardized z scores. The source of these data was the WorldClim database[57], which have been averaged from their original 1 km resolution to a 10 km resolution for this analysis using the WaterWorld platform[58]. WorldClim provides climatology of monthly mean precipitation and rainfall from interpolated station data over the period 1950–2000. In our analysis we look at average long-term climate and long-term DHS DD incidence with a focus on spatial patterns rather than temporal ones. We are not studying how weather conditions affect DD rates on a specific date but how geographical differences in climate and land cover and use affect geographical differences in long-term DD incidence. If the 2000–2012 period does not depart from the long-term climate in a region then our analysis would be reasonable. Where countries have seen unusual drought or flood conditions during 2000–2012 there would be greater uncertainty in our associations between climate and DD.

*Watershed conditions—influence of upstream human activity and influence of upstream tree cover*: we develop two measures of watershed condition and their influence on water quality, using metrics developed for WaterWorld[58, 59]. These metrics estimate the percent of water at any point in a river network that fell as rain on any defined land cover or use category[59]. WaterWorld accounts for the spatial distributions of land cover or use and rainfall, then routes water downstream to understand the build up and dilution of potential water contamination. The rainfall data set used is WorldClim[57] and the flow network is HydroSHEDS[60]. Water balance is calculated from WorldClim rainfall minus actual evapotranspiration calculated from a 10-year climatology of the MODIS actual evapotranspiration product[61]. The first variable measures the potential influence of upstream people and livestock on water quality. We define two variables as potential sources of DD: human populations using Landscan data[62] and cattle ranches[63]. For each pixel, the percent of water falling as rain on upstream areas with human population > 0 or cattle headcount > 0 is calculated[62, 63]. The area of human impact depends on the

population in each 10 km × 10 km pixel (each human is assumed to contaminate 3.65 m² of ground per year). All pasture land is assumed polluted as long as the cattle headcount is > 0[64]. This is routed downstream as contaminated runoff. Index values of 0 mean no presence of human or livestock inputs or no water. A value of 100% indicates that all water in the current pixel fell as rain on land used for human population or livestock. The second variable measures the potential influence of upstream tree cover on water quality. Similar to human activity, for each pixel, the percent of water falling as rain on tree covered areas from MODIS data[65] is calculated and cumulated downstream as a percentage of the total cumulated water[58]. Values of 0 for this variable mean that there is no presence of local and upstream trees or no water. These variables can be calculated for any watershed using the freely available WaterWorld tool (www.policysupport.org/waterworld).

**Analyses**. We analyzed our data set using mixed effects hierarchical logit models. These models provide an appropriate framework to study nested data sets and account for the effect of covariates measured at different levels of a hierarchical structure, correcting the biases in parameter estimates resulting from clustering and providing adjusted standard errors[66]. Our data set has three levels: individual characteristics (level 1), household characteristics (level 2), and cluster characteristics (group of households in the survey design, level 3), and therefore we use a three-level nested model which includes random intercepts at the household and cluster level. The model assumes households add a random effect to the probability of DD, and these effects vary from household to household. Since households are nested within clusters the model adds a second random term which varies by cluster. We did not add country random effects since we found a low intra-class correlation coefficient of 0.01 at this level, lower than the coefficients at the household and cluster levels, 0.18 and 0.05, respectively. We would therefore expect a low effect of adding country level random effects on the standard errors of the model coefficients. To confirm this statement, after estimating the model we checked the relationship between the country mean of the cluster-level random effect and the country mean of the predicted values of DD to verify if the cluster level random effect is effectively controlling for country-level variation in DD, and we did find a strong positive relationship (Supplementary Fig. 2).

To specify our basic model, we observe $y_{ijk}$, a binary diarrhea variable for child $i$ in household $j$ in cluster $k$. We define the probability of diarrhea equal to 1 as $p_{ijk} = \Pr(y_{ijk} = 1)$ and let $p_{ijk}$ be modeled using a logit link function. The three level model can be written as:

$$
\begin{aligned}
\log\left[p_{ijk}/(1 - p_{ijk})\right] = {} & \beta_0 + \beta_1 \mathrm{Age}_{ijk} \\
& + \beta_2 \mathrm{Wealth}_{jk} \\
& + \beta_3 \mathrm{Education}_{jk} \\
& + \beta_4 \mathrm{Improved\ sanitation}_{jk} \\
& + \beta_5 \mathrm{Improved\ water}_{jk} \\
& + \beta_6 \mathrm{Precipitation}_{t-1_k} \\
& + \beta_7 \left(\mathrm{Precip}_t - \mathrm{Precip}_{t-1}\right)_k \\
& + \beta_8 \mathrm{Temperature}_{t-1_k} \\
& + \beta_9 \left(\mathrm{Temp}_t - \mathrm{Temp}_{t-1}\right)_k \\
& + \beta_{10} \mathrm{Human\ activity\ upstream}_k \\
& + \beta_{11} \mathrm{Tree\ cover\ upstream}_k \\
& + v_{0k} + u_{0jk}
\end{aligned}
$$

where

$v_{0k}$ = cluster-level random intercept, independent across clusters
$v_{0k} \sim N(0, \sigma_k)$, $\sigma_k$ is the residual between-cluster variance
$u_{0jk}$ = household-level random intercept, independent across households, within clusters
$u_{0jk} \sim N(0, \sigma_{jk})$, $\sigma_{jk}$ is the residual between-household, within-cluster variance

We also explore an alternative specification including age as binary categorical variables and upstream tree cover as binary variables representing quartiles, which supports the results of our main comparison between urban and rural households (Supplementary Fig. 3). We used Stata 14's *melogit* to estimate our models, which provides large sample approximations for our confidence intervals estimates.

Supplementary Table 7 presents the pairwise linear correlations between the variables in our model. All correlation coefficients are lower than 0.4, which indicate low to moderate correlations. We could expect upstream human activity and tree cover to be negatively correlated for a particular region, which could bias the results of our model. We found a low correlation coefficient of −0.03 between these two variables, which should not be a concern for our analysis.

In nonlinear models the marginal effects differ from the estimated coefficient as these depend on the values of the other explanatory variables, and in our case, also depend on the estimated random effects. The interpretation of marginal effects differs between binary and continuous variables. The socioeconomic variables in our model are binary so the marginal effect corresponds to changes in each of these variables from 0 to 1, i.e., low to high levels or unimproved to improved states. On the other hand, the marginal effect of a continuous independent variable such as the upstream tree cover variable is the instantaneous rate of change, i.e., the change

in the outcome variable given small changes in the independent variable (close to zero). We use the latter measure to approximate changes in the probability of DD for three scenarios of change in upstream tree cover influence: 10, 20, and 30% increases. We used Stata's margins command with the atmeans option to calculate the marginal effects.

**Data availability**. The code and environmental data that support the findings of this study are available from the corresponding author upon request. The primary health data used in this analysis are available from http://dhsprogram.com/Data/.

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

## Acknowledgements

This work was supported by the National Socio-Environmental Synthesis Center (SESYNC) under funding from the National Science Foundation DBI-1052875, the Gordon and Betty Moore Foundation and The Rockefeller Foundation as part of the Health & Ecosystems: Analysis of Linkages (HEAL) program, and the Luc Hoffmann Institute at WWF International under funding provided by the Mava Foundation. We thank Sam Myers, Anila Jacob, Mary Shelley, Jon Kramer, Louise Gallagher, David Tickner, Sarah Gallalee, and University of Vermont's Spatial Analysis Lab for their valuable inputs.

## Author contributions

T.H.R. and B.F. directed the SESYNC group that led to this article. T.H.R., B.F., D.H., and A.E. designed the study. D.H. and A.E. compiled data sets and ran statistical analyses. D.H. wrote the manuscript. K.J., C.D.G., A.P., M.M. and T.T. advised on proper use of data, methods and literature. M.M. generated hydrological watershed data and advised on proper use of data. All authors contributed to analyses, interpretation and writing of the final manuscript.

## Additional information

**Competing interests:** The authors declare no competing financial interests.

