## [Peer Review File · Nature Communications]

Reviewers' Comments:

Reviewer #1 (Remarks to the Author)

This is an excellent paper. The methods are sound and novel, and the results important. Most of my comments are relatively minor.

-Rob McDonald

Overall issues:

1.) At times the paper is a bit loose with its language, implying that their statistical result is clearly related to ecosystem service provision. Another, equally valid interpretation of many of their explanatory variables is that they are correlated with increased anthropogenic pollution. So, there is a complex of effects: natural habitat clearing reduces ecosystem service provision, but goes along with increasing human population density and activity, and hence increase anthropogenic pollution.

2.) See my notes below about the pitfalls of depicting these results as green versus grey, rather than green and gray as part of a blended solution. This is mostly a stylistic issue, and can be changed with minor rewriting of some sentences.

3.) I would like to see more info on the correlation and relationship between tree cover and human activity. To their credit, the authors have already done some analyses of this, but it is a bit buried right now in the supplementary material. See my comments below.

4.) The paper needs to discuss more clearly the difference between improved and unimproved sources, perhaps partitioning on this variable (recognizing that space is tight, that extra material could be in the supplement). Moreover, to help readers understand the different results for urban versus rural dwellers, it is worthwhile describing some of the literature that shows that urban dwellers often source water farther from where they live, and often upstream from the city or in a different basin entirely. This literature, of course, only is a further explanation of why urban dwellers would be much less affected by watershed condition, as analyzed in this study, which looked at the watershed upstream of individuals rather than their water sources per se.

Title: Nice and clear description of main finding.

Abstract: Good clear summary of the motivation of the report, its methods, and its findings. Could potentially benefit from a mention of one major difference between urban dwellers and rural dwellers: the former often source from watersheds far from the city, often across watershed boundaries, while the latter are much more likely to be sourcing water locally. This is tangentially referred to in the abstract but could be more clearly stated.

Also, I worry the phrasing of the abstract would imply the natural infrastructure can or should replace grey infrastructure like treatment plants. While the analysis in this paper implies natural infrastructure supplies benefits of similar magnitude to grey infrastructure, would be good to be clear that you are making that relative comparison just to make the magnitude clear, not implying that (say) protecting forest replaces the goal of getting sanitation or improved water sources to rural dwellers.

Introduction:

Lines 84-85: Please give an estimate of the proportion of people in the developing world that rely on unimproved surface water sources. This is only very imprecisely known in the literature, I acknowledge, but this seems like a crucial factoid for interpreting this papers results.

Lines 104-105: I'd suggest adding a sentence here with the breakdown in the sample that was rural/urban, as well as the proportion of the sample that had unimproved versus improved supplies. I know that may be elsewhere in the methods, but seems good context for interpreting these results.

Results:

Lines 139: Would be good to include a sentence on the correlation between forest cover and human activity variables, as I am presuming they are negatively correlated to each other. Indeed, if this negative correlation is strong I would be worried about your statistical ability to separate these two effects. Might be good to have a scatterplot or some other graphic of this relationship in the online info (given the many data points, perhaps a scatterplot isn't helpful for visualizing the relationship, but something then like a 2-D kernel density estimation, just to see the relationship between the two variables).

Results 153: In addition to the other things that vary with urban vs. rural status, worth noting here that urban locations often source from far from city centers (with the significant exception of many in urban slums, which may predominate in the DHS data), and often in different watersheds than where the individual is located. Thus, even if there was a relationship between watershed condition and health, it may be unapparent the way the current analysis was conducted, looking at the locations of individuals.

Figure 2- Love this figure, a nice clear statement of the model results. I wish more papers did this!

This may be covered in the methods elsewhere, but how are you dealing with interactions among variables here? One in particular seems key to me: You would expect the human activity and forest cover variables to be important when a household does not have improved water, and of little importance when a household has improved water. How was that interaction dealt with?

Line 263- "...perhaps highlighting the importance..." I confess I don't understand the point being made. Why would improved water be less important in regions with less infrastructure? Clear water is clean water, right? Or are these sources not really "improved"?

Discussion:

Lines 323- I worry about this explicit contrasting of green versus grey here, since it could make it seem like the conservation community is saying forest protection would REPLACE building treatment plants and sanitation. I would encourage the authors to recast this language (with the same statistical factoid) as showing that protecting forests can be a useful part of how countries try to achieve health (in other words, both green and grey will be needed).

Here, there is a mention of fitting the model for watersheds with both high and low levels of human activity, to account for the correlation between the human activity and forest cover variables. This is a really interesting analysis, and deserves to be treated in the results, rather than just mentioned in passing in the discussion.

Methods:

Watershed conditions paragraph: Might be worth writing out that you used things like the Landscan database, rather than making the reader follow the citation to the bibliography. What is the nominal year of the population data in the Landscan database, and does that accord with the years for which you have the DHS data?

In addition to splitting the data on urban/rural to see if that effects results, it seems to me like splitting the sample on improved/not improved would perhaps be useful, since that is a variable that has clear hydrologic and health importance. Obviously, the urban households are much more

likely to be improved, so I understand this may not change the results much, but still seems like it would be worthwhile to examine this.

Supplement:

I find table SI2 helpful. It might be worth transposing the table, and then showing some of the other variables average values in urban and rural.

SI3 is a useful figure, that goes some way toward answering the questions I posed in my comments on the Results, re: the correlation between human activity and forest cover. Even if there is no room for this figure to be actually in the body text, this deserves some discussion in the Results, and you can direct readers to the supplement for more detail.

Reviewer #2 (Remarks to the Author)

This manuscript quantifies the relationship between watershed condition (tree cover and human and livestock population density) upstream of water sources and incidence of childhood diarrheal disease in 35 developing countries. The results indicate a positive association between watershed condition (more trees, lower density) and incidence for rural but not urban water sources. This is a very interesting finding that deserves to be published. The study contributes to the growing body of literature about the relationship between ecosystem condition and human health. However, I suggest that the authors tighten the text in several places, be more precise in terminology and interpretation, and frame the study in the context of the health outcome.

General comments:

1. Title – Please consider changing the title to reflect what the study is actually about – “childhood diarrheal disease” rather than the too-broad term “children’s health” and “tree cover and population density” rather than the too-broad term “watershed condition”.

2. A more direct approach to test the relationship between watershed properties and diarrhea-causing pathogens is through water quality measurements. Obviously, water quality measurements are very sparse in the countries included in the study. Are there any direct water quality measurements that would lend support to the analysis from these countries or other places? It seems remiss not to comment on studies of in situ water quality measurements.

3. This comment might just be by personal reaction, but I find the framing of the study around the largely-hand waving issue of ecosystems and human health to get off on the wrong foot. It raises suspicions that the authors are out to “prove” that ecosystem degradation necessarily has a negative impact on human health. I would prefer that the authors stick to the question at hand – are tree cover and population density in watersheds associated with incidence of childhood diarrheal disease? It is more powerful to stick to the evidence on this question rather than editorialize about generic relationships between ecosystems and health (again, this just might be my reaction).

Specific comments:

4. Lines 16-18: I suggest deleting the first two sentences of the abstract. All terrestrial places on Earth are part of a watershed, so it is meaningless to say watersheds are at particular risk. “Rapidly degrading” is equally vague. Deforested? Contaminated? I would prefer to get to the real issue which is about diarrheal disease and water quality.

5. Line 19: What does “watershed degradation” mean specifically? There are many kinds of degradation – loss of vegetation, fertilizer runoff, point sources for pollution etc.

6. Lines 21-22: What does “watershed condition” mean specifically? I think it means tree cover

and population and livestock density.

7. Lines 29-30: I think the statement that intact ecosystems are associated with better health outcomes is a stretch. Could tree plantations (non-intact ecosystem) provide a similar benefit?

8. Lines 59-63: I would favor deleting the first paragraph because it sounds like you are biased in trying to prove that natural ecosystems are good for health (I'm not implying that you are biased but some might interpret that way). Better to bend over backwards to avoid that perception.

9. Line 111: Do you mean DENSITY of human and livestock population upstream? How did you combine human and livestock into a single measure? Why didn't you keep human and livestock separate to see if there is an improvement in water quality by reducing one or the other independently?

10. Line 118 – 119: I'm not sure I understand the three levels. Are there multiple individuals (children under 5) from the same household in the data set?

11. Line 141: More information on the distribution of sizes of the upstream watershed would be useful. Are all locations in the watershed equal in terms of contributing to downstream water quality? I'm not suggesting new analysis, but it seems logical that places closer to the water source would have more impact. Or does the HydroSHEDS method account for distance from water source? I think so but I'm not sure. Can you clarify?

Line 150: I would expect that precipitation would be positively correlated with probability of diarrhea since more wastes wash downstream. This study finds otherwise. Does literature relating in situ water quality measurements to precipitation support your results?

12. Line 211: I'd like to know the baseline probability of experiencing diarrhea. Is a 5 percent reduction a lot or a little compared to the baseline?

13. Line 369: "Unpacked" seems like a colloquial terminology for use in a scientific manuscript.

14. Line 401 and 439: There seems to be a time mismatch between the 2 week recall period for diarrheal disease and the monthly mean temperature and precipitation data. A finer resolution climate data would involve a lot of work to extract daily data so it probably is not worth it. How much variability is there in temperature and precipitation within the month and is the mean representative of temperature and precipitation in the two week recall? This issue seems particularly pertinent for precipitation where a rainfall event after the two week period could bias the result.

15. Line 422 and Fig S17 and associated text: For water sources from tube well and borehole, it seems you are assuming that the watershed for surface water is the same as for groundwater. It would be difficult to know the groundwater source area, but the assumption does not seem valid.

16. Line 499: What is the data source for land cover and tree cover?

17. S19: I trust you tested for co-linearity of variables in the model but I don't see that stated (or maybe I missed it).

Reviewer #3 (Remarks to the Author)

This manuscript concerns the potential impacts of watershed conditions, including natural and anthropogenic influences, on diarrheal diseases. This is an important topic given the burden of diarrheal diseases. It is also timely, given the recent planetary health report calling for greater investigation of relationships between ecological processes and human health. However, I have several major concerns about the execution of this analysis, detailed below.

The authors merge multiple datasets including health (reported diarrhea from DHS), climatological (temperature and precipitation), demographic and ecological (watershed conditions) data. While the authors describe the spatial resolution of the health data, it is unclear to what extent the spatial resolution of the merged datasets correspond. This is a particular concern for the DHS data, as spatial uncertainty is intentionally added to the data in order to protect participant privacy. Given this spatial uncertainty, to what extent are survey respondents potentially misclassified as to their watershed? Similarly, what is the potential for misclassification of human or bovine presence in a watershed, given the datasets used to define these variables?

Meteorological data from 1950 to 2000 are used and applied to health data from 2001 to 2012. This decision is not justified by the authors and is puzzling given the number of meteorological products available for the study period.

The definition of watersheds is not well described and largely included in the references. Given the multi-disciplinary approach of this manuscript, at least a cursory description of watershed delimitation should be provided.

There is a lack of detail and justification of modeling decisions, particularly concerning the covariates in the model. For example, monthly mean temperature and precipitation were used but health data are at two-week temporal scale. Temporal lags between the climate and health data were not discussed here despite extensive work in this area. Another example: age in months appears to be treated as a continuous variable assuming a linear relationship between age and diarrheal disease. A categorical definition might be more appropriate and avoid assumptions of linearity. Similarly, human activity and tree cover appear to be modeled as continuous variables without discussion of the shape of the exposure response curves. Given that these are the exposures of interest, a more thoughtful consideration of the exposure response relationship is warranted (e.g. through sensitivity analyses that examine the relationship in quartiles).

Reviewers' comments

Reviewer #1 (Remarks to the Author):

This is an excellent paper. The methods are sound and novel, and the results important. Most of my comments are relatively minor.

Overall issues:

1.) At times the paper is a bit loose with its language, implying that their statistical result is clearly related to ecosystem service provision. Another, equally valid interpretation of many of their explanatory variables is that they are correlated with increased anthropogenic pollution. So, there is a complex of effects: natural habitat clearing reduces ecosystem service provision, but goes along with increasing human population density and activity, and hence increase anthropogenic pollution.

Thanks for your comment. In this version of the paper we are more explicit about acknowledging that multiple mechanisms are possible and that even though we are looking at associations between variables in the paper, we tried to test and rule out competing mechanism by using key sample splits across areas with high and low upstream human pressures and this result is now in the main text (Figure 5)..

2.) See my notes below about the pitfalls of depicting these results as green versus grey, rather than green and gray as part of a blended solution. This is mostly a stylistic issue, and can be changed with minor rewriting of some sentences.

Thank you for this comment. We have edited the text to be more careful about the not presenting our main results as green vs gray. We do highlight the value provided by sanitation and other forms of human capital. But we do find that in a setting where those elements are not present green could help fill the gap.

3.) I would like to see more info on the correlation and relationship between tree cover and human activity. To their credit, the authors have already done some analyses of this, but it is a bit buried right now in the supplementary material. See my comments below.

Thank you for this comment. We included additional text and results to the main text describing these correlations and how we try to address them (Line 327, Figure 5)

4.) The paper needs to discuss more clearly the difference between improved and unimproved sources, perhaps partitioning on this variable (recognizing that space is tight, that extra material could be in the supplement). Moreover, to help readers understand the different results for urban versus rural dwellers, it is worthwhile describing some of the literature that shows that urban dwellers often source water farther from where they live, and often upstream from the city or in a different basin entirely. This literature, of course, only is a further explanation of why urban dwellers would be much less affected by watershed condition, as analyzed in this study, which looked at the watershed upstream of individuals rather than their water sources per se.

Thanks. We added additional citations describing the differences between urban and rural dwellers (lines 84-91)

Title: Nice and clear description of main finding.

Abstract: Good clear summary of the motivation of the report, its methods, and its findings. Could potentially benefit from a mention of one major difference between urban dwellers and rural dwellers: the former often

source from watersheds far from the city, often across watershed boundaries, while the latter are much more likely to be sourcing water locally. This is tangentially referred to in the abstract but could be more clearly stated.

Thanks for this comment. Due to space constraints this could not be added to the abstract but we make the point about urban/rural differences more explicit in the introduction (lines 84-91)

Also, I worry the phrasing of the abstract would imply the natural infrastructure can or should replace grey infrastructure like treatment plants. While the analysis in this paper implies natural infrastructure supplies benefits of similar magnitude to grey infrastructure, would be good to be clear that you are making that relative comparison just to make the magnitude clear, not implying that (say) protecting forest replaces the goal of getting sanitation or improved water sources to rural dwellers.

Thanks for this comment. We revised the main text to make sure we are not implying that people should not invest in human capital or manmade infrastructure because of the upstream presence of tree cover.

Introduction:

Lines 84-85: Please give an estimate of the proportion of people in the developing world that rely on unimproved surface water sources. This is only very imprecisely known in the literature, I acknowledge, but this seems like a crucial factoid for interpreting this papers results.

Thanks for this suggestion. We added additional citations to justify our urban vs rural comparison (line 84)

Lines 104-105: I'd suggest adding a sentence here with the breakdown in the sample that was rural/urban, as well as the proportion of the sample that had unimproved versus improved supplies. I know that may be elsewhere in the methods, but seems good context for interpreting these results.

We added this information to the main text (lines 156-157)

Results:

Lines 139: Would be good to include a sentence on the correlation between forest cover and human activity variables, as I am presuming they are negatively correlated to each other. Indeed, if this negative correlation is strong I would be worried about your statistical ability to separate these two effects. Might be good to have a scatterplot or some other graphic of this relationship in the online info (given the many data points, perhaps a scatterplot isn't helpful for visualizing the relationship, but something then like a 2-D kernel density estimation, just to see the relationship between the two variables).

Thanks for this suggestions. This is an important point that we have included in our discussion and methods sections. For our sample of clusters we calculated and discussed the pairwise correlation between upstream tree cover and upstream human activity (lines 133-134, 532-538)

Results 153: In addition to the other things that vary with urban vs. rural status, worth noting here that urban locations often source from far from city centers (with the significant exception of many in urban slums, which may predominate in the DHS data), and often in different watersheds than where the individual is located. Thus, even if there was a relationship between watershed condition and health, it may be unapparent the way the current analysis was conducted, looking at the locations of individuals.

Thanks for your comment. This is a great point consistent with the fact that we are not finding significant impacts of watershed conditions in urban areas.

Figure 2- Love this figure, a nice clear statement of the model results. I wish more papers did this!

This may be covered in the methods elsewhere, but how are you dealing with interactions among variables here? One in particular seems key to me: You would expect the human activity and forest cover variables to be important when a household does not have improved water, and of little importance when a household has improved water. How was that interaction dealt with?

Thanks for this comment. This is a key robustness check and so we added additional analysis to the main text where we split our sample into households with and without improved water to test the model (and the effect of tree cover) across the two groups. We find that the tree cover variables is significant in the group without improved water (Figure 4).

Line 263- "...perhaps highlighting the importance..." I confess I don't understand the point being made. Why would improved water be less important in regions with less infrastructure? Clear water is clean water, right? Or are these sources not really "improved"?

Thank you for your question. The point we are trying to make is that where there is less gray infrastructure the natural conditions might play a relatively larger role as a determinant of health outcomes. This made in line 231, Figure 4.

Discussion:

Lines 323- I worry about this explicit contrasting of green versus grey here, since it could make it seem like the conservation community is saying forest protection would REPLACE building treatment plants and sanitation. I would encourage the authors to recast this language (with the same statistical factoid) as showing that protecting forests can be a useful part of how countries try to achieve health (in other words, both green and grey will be needed).

We agree on the importance of being cautious about how we frame the results and the comparison between green and gray. We have rephrased this paragraph.

Here, there is a mention of fitting the model for watersheds with both high and low levels of human activity, to account for the correlation between the human activity and forest cover variables. This is a really interesting analysis, and deserves to be treated in the results, rather than just mentioned in passing in the discussion.

Thanks for this suggestion. We moved these results from the supplementary material to the main text (Figure 5).

Methods:

Watershed conditions paragraph: Might be worth writing out that you used things like the Landscan database, rather than making the reader follow the citation to the bibliography. What is the nominal year of the population data in the Landscan database, and does that accord with the years for which you have the DHS data?

The year of the population data is 2007 which overlaps with the period under study. We added these information to the methods section (line 493).

In addition to splitting the data on urban/rural to see if that effects results, it seems to me like splitting the sample on improved/not improved would perhaps be useful, since that is a variable that has clear hydrologic and health importance. Obviously, the urban households are much more likely to be improved, so I understand this may not change the results much, but still seems like it would be worthwhile to examine this.

We agree this is an important robustness check and we have added this sample split analysis to the main text (Figure 4).

Supplement:

I find table SI2 helpful. It might be worth transposing the table, and then showing some of the other variables average values in urban and rural.

Thanks for this suggestion. We have transposed the table and added the other variables in the model (Table SI.2)

SI3 is a useful figure that goes some way toward answering the questions I posed in my comments on the Results, re: the correlation between human activity and forest cover. Even if there is no room for this figure to be actually in the body text, this deserves some discussion in the Results, and you can direct readers to the supplement for more detail.

We have moved this result from the supplementary material to the main text (Figure 5).

Reviewer #2 (Remarks to the Author):

This manuscript quantifies the relationship between watershed condition (tree cover and human and livestock population density) upstream of water sources and incidence of childhood diarrheal disease in 35 developing countries. The results indicate a positive association between watershed condition (more trees, lower density) and incidence for rural but not urban water sources. This is a very interesting finding that deserves to be published. The study contributes to the growing body of literature about the relationship between ecosystem condition and human health. However, I suggest that the authors tighten the text in several places, be more precise in terminology and interpretation, and frame the study in the context of the health outcome.

General comments:

1. Title – Please consider changing the title to reflect what the study is actually about – “childhood diarrheal disease” rather than the too-broad term “children’s health” and “tree cover and population density” rather than the too-broad term “watershed condition”.

Thanks for this suggestion. We kept the title somewhat broad but tried to reflect better our main finding for rural households, while presenting more clearly our result for upstream tree cover in the abstract (line 20)

2. A more direct approach to test the relationship between watershed properties and diarrhea-causing pathogens is through water quality measurements. Obviously, water quality measurements are very sparse in the countries included in the study. Are there any direct water quality measurements that would lend support to the analysis from these countries or other places? It seems remiss not to comment on studies of in situ water quality measurements.

While we don't have water quality measurements (would not be feasible to collect this data for all points in our sample) we added a citation of a study linking water quality measures and diarrhea and added more information on the potential mechanisms between watershed conditions and health (lines 68-73).

3. This comment might just be by personal reaction, but I find the framing of the study around the largely-hand waving issue of ecosystems and human health to get off on the wrong foot. It raises suspicions that the authors are out to “prove” that ecosystem degradation necessarily has a negative impact on human health. I would prefer that the authors stick to the question at hand – are tree cover and population density in watersheds associated with incidence of childhood diarrheal disease? It is more powerful to stick to the evidence on this question rather than editorialize about generic relationships between ecosystems and health (again, this just might be my reaction).

Thanks for this comment. We modified the introduction to make it more succinct and move quicker to the research questions we are interested.

Specific comments:

4. Lines 16-18: I suggest deleting the first two sentences of the abstract. All terrestrial places on Earth are part of a watershed, so it is meaningless to say watersheds are at particular risk. "Rapidly degrading" is equally vague. Deforested? Contaminated? I would prefer to get to the real issue which is about diarrheal disease and water quality.

Related to the previous comment. We re-wrote the introduction to be clearer about the objectives of our research.

5. Line 19: What does "watershed degradation" mean specifically? There are many kinds of degradation – loss of vegetation, fertilizer runoff, point sources for pollution etc.

6. Lines 21-22: What does "watershed condition" mean specifically? I think it means tree cover and population and livestock density.

We edited the main text to be more explicit about the upstream population/livestock measure as well as the upstream tree cover measure (lines 119-126)

7. Lines 29-30: I think the statement that intact ecosystems are associated with better health outcomes is a stretch. Could tree plantations (non-intact ecosystem) provide a similar benefit?

Thanks for this comment. Tree plantations could provide a similar benefits are long as these are not grazed or populated. We've removed the reference to intact ecosystems.

8. Lines 59-63: I would favor deleting the first paragraph because it sounds like you are biased in trying to prove that natural ecosystems are good for health (I'm not implying that you are biased but some might interpret that way). Better to bend over backwards to avoid that perception.

Thanks for this suggestion. We deleted the paragraph from the introduction.

9. Line 111: Do you mean DENSITY of human and livestock population upstream? How did you combine human and livestock into a single measure? Why didn't you keep human and livestock separate to see if there is an improvement in water quality by reducing one or the other independently?

We do not measure upstream density of people and livestock per se. The details about how this variable is defined is in the Methods section (line 482). This is a hydrologic measure of the influence of upstream populations and livestock on water downstream as a single indicator of a potential sources of diarrheal disease driven by human activities.

10. Line 118 – 119: I'm not sure I understand the three levels. Are there multiple individuals (children under 5) from the same household in the data set?

That would be correct. Children (level 1) are grouped into households (level 2), and households are grouped into clusters (level 3). We rewrote this sentence to make the point clear (line 515).

11. Line 141: More information on the distribution of sizes of the upstream watershed would be useful. Are all locations in the watershed equal in terms of contributing to downstream water quality? I'm not suggesting new analysis, but it seems logical that places closer to the water source would have more impact. Or does the HydroSHEDS method account for distance from water source? I think so but I'm not sure. Can you clarify?

Thanks for your questions. Our measures are indeed weighted by hydrological influence (as opposed to averages across watersheds). Full details in Methods, line 482.

Line 150: I would expect that precipitation would be positively correlated with probability of diarrhea since more wastes wash downstream. This study finds otherwise. Does literature relating in situ water quality measurements to precipitation support your results?

The results show that conditional on the other factors precipitation would reduce the probability of diarrhea. This effect is described in the literature as a dilution effect. Areas with low precipitation levels will end up with high concentrations of pollutants per unit water. Areas with more water will have these diluted below safe limits.

12. Line 211: I'd like to know the baseline probability of experiencing diarrhea. Is a 5 percent reduction a lot or a little compared to the baseline?

The baseline measures are presented in the same paragraph as the % reductions in this version of the paper (lines 212 and 220)

13. Line 369: "Unpacked" seems like a colloquial terminology for use in a scientific manuscript.

Thanks for this suggestion. The word was deleted from the manuscript.

14. Line 401 and 439: There seems to be a time mismatch between the 2 week recall period for diarrheal disease and the monthly mean temperature and precipitation data. A finer resolution climate data would involve a lot of work to extract daily data so it probably is not worth it. How much variability is there in temperature and precipitation within the month and is the mean representative of temperature and precipitation in the two week recall? This issue seems particularly pertinent for precipitation where a rainfall event after the two week period could bias the result.

Thanks for this suggestion. We ran our models again using lagged temperature and precipitation as well as the differences between current and previous month in an effort to address these temporal concerns. This does not appear to affect the main findings of the paper (Figures 2, 4 and 5)

15. Line 422 and Fig S17 and associated text: For water sources from tube well and borehole, it seems you are assuming that the watershed for surface water is the same as for groundwater. It would be difficult to know the groundwater source area, but the assumption does not seem valid.

Thanks for your comment. The assumption we previously made is that ground water is less directly impacted by land use compared to surface water and therefore could be considered 'improved'. However, we acknowledge that a better assumption could be to define our improved water variable as only piped sources. We redefined the improved water variable according to this to use a stricter definition and re-ran our analysis based on this (see table SI.7).

16. Line 499: What is the data source for land cover and tree cover?

This information is currently in the Methods section (lines 482-506). Sources are MODIS, Landsat, Hydrosheds and Robinson et al 2013 (citation 64).

17. S19: I trust you tested for co-linearity of variables in the model but I don't see that stated (or maybe I missed it).

There is correlation among some of these variables, the highest between education and improved toilet with a correlation coefficient of 0.33. The rest of the variables have lower correlation coefficients that are considered either low or moderate correlations.

Reviewer #3 (Remarks to the Author):

This manuscript concerns the potential impacts of watershed conditions, including natural and anthropogenic influences, on diarrheal diseases. This is an important topic given the burden of diarrheal diseases. It is also timely, given the recent planetary health report calling for greater investigation of relationships between ecological processes and human health. However, I have several major concerns about the execution of this analysis, detailed below.

The authors merge multiple datasets including health (reported diarrhea from DHS), climatological (temperature and precipitation), demographic and ecological (watershed conditions) data. While the authors describe the spatial resolution of the health data, it is unclear to what extent the spatial resolution of the merged datasets correspond. This is a particular concern for the DHS data, as spatial uncertainty is intentionally added to the data in order to protect participant privacy. Given this spatial uncertainty, to what extent are survey respondents potentially misclassified as to their watershed? Similarly, what is the potential for misclassification of human or bovine presence in a watershed, given the datasets used to define these variables?

Thanks for your comment. Our strategy for merging health and environmental data is based on defining a grid resolution for the environmental data which is large enough (10km) to encompass all possible displacement of the DHS health data coordinates. We provide an explanation in the Methods and Materials section (lines 426-435)

Meteorological data from 1950 to 2000 are used and applied to health data from 2001 to 2012. This decision is not justified by the authors and is puzzling given the number of meteorological products available for the study period.

Thank you for this comment. In our analysis we attempt to compare average long term climate with average long term diarrheal disease rates and doing this geographically (spatially, not temporally). Our study is not about comparing diarrheal disease rates with the weather conditions around those rates. If the 2001-2012 period does not depart from the long term climate in a region then our analysis would be reasonable. For most places that will be true, especially as our main focus is on effects of tree cover, not climate

There is now a new version of WorldClim that is 1970-2000 but there is no global high resolution gridded data for 2001-2012 (the CRU UEA 10km data goes to 2002 only). The only way to do this with 2001-2012 data would be to use reanalysis data (modelled climate coupled with measured) and the best possible resolution to our knowledge would be >30km and more likely 50km resolution or worse. This is much greater scale than the variability in tree cover and would render the analysis much weaker. Station data or coarse resolution (30km products) will not do given the way local rainfall close to clusters has a significant impact on delivery of water and water quality at the cluster. We need long term monthly averages not month by month data as we do not want to change our focus to a longitudinal analysis (because then we would need time series for all the other factors that might also change month to month 2001-2012 (cattle, people, tree cover)).

The definition of watersheds is not well described and largely included in the references. Given the multi-disciplinary approach of this manuscript, at least a cursory description of watershed delimitation should be provided.

Thanks for this comment. The metrics we are calculating actually are derived from an existing gridded dataset, produced by Mark Mulligan, which identifies upstream areas in order to calculate the metrics. The full explanation of how these variables are defined is in Methods (line 482), which are hydrologic measures averaged at a 10km grid resolution around the location of the household clusters based on river networks.

There is a lack of detail and justification of modeling decisions, particularly concerning the covariates in the model. For example, monthly mean temperature and precipitation were used but health data are at two-week temporal scale. Temporal lags between the climate and health data were not discussed here despite extensive work in this area.

Thanks for this comment. All variables are defined in Methods (lines 436-506) We have added lagged climate variables to our model following your suggestion and existing literature.

Another example: age in months appears to be treated as a continuous variable assuming a linear relationship between age and diarrheal disease. A categorical definition might be more appropriate and avoid assumptions of linearity.

Thanks for this suggestion. We have added a robustness check on this point by running a model with categorical variables for age (Figure 5 in the main text) showing a nonlinear trend established in the literature where the probability of diarrhea increases and peaks at the second year of age (relative to the first year of age) and then decreases as the child gets older.

Similarly, human activity and tree cover appear to be modeled as continuous variables without discussion of the shape of the exposure response curves. Given that these are the exposures of interest, a more thoughtful consideration of the exposure response relationship is warranted (e.g. through sensitivity analyses that examine the relationship in quartiles).

Thanks for this suggestion. We have split our upstream tree cover variable in quartiles to test which levels generate a significant effect on the probability of diarrhea, relative to the lowest quartile. This approach also allows us to compare directly the magnitude of the effect of our binary tree variables with the binary socioeconomic variables also included in the model (Figure 5).

Reviewers' Comments:

Reviewer #2:

Remarks to the Author:

The manuscript is much improved in response to reviewer comments. I am mostly satisfied with the responses and revisions. A few questions remain, which I am happy to leave to the judgment of the authors and editor:

- I still don't understand why the authors did not include humans and livestock as separate variables rather than combining into a single "human activity" variable. It seems potentially policy-relevant to assess whether the impact on downstream diarrheal disease is related to humans or livestock or both.

- It would be useful to show in a supplemental table the correlation between variables used in the model to assure that there is not a problem of co-linearity.

- I still think the title would be more appropriate if it said "childhood diarrheal disease" rather than the broad term "children's health"

The paper describes an important result that deserves publication.

Reviewer #3:

Remarks to the Author:

The authors have revised the manuscript and improved it in many ways. I do have one remaining concern that was raised in my initial review and not fully addressed.

In the main model (results shown in Fig 2) age, human activity and tree cover appear to have been modeled as continuous variables. I suggested they authors conduct a sensitivity analysis to evaluate the appropriateness of this modeling decision. In their revision, the authors have done this, but only for a subset of the data: Figure 5 presents estimates for age and upstream tree cover as categorical variables, but only for rural populations divided into populations with high vs. low upstream human activity. This is an interesting figure and allows exploration of interactions between upstream human activity and tree cover. However, as the marginal effects are based on the full model, the sensitivity analysis should be presented for the same data and model. Depending on the results of this analysis, the findings could be presented in the Supplementary Information at your discretion. But, in the absence of this information supporting the modeling of your exposure variables as continuous, I don't think the marginal effects estimates, which appear to be the main findings of the paper, are justified.

Minor comment.

The table titles and notes in the Supplementary Appendix could be revised to describe the information presented more thoroughly. It would be nice, for example, to describe in a footnote what information is provided in the parentheses in tables S1.1 and S1.3. Good epidemiological practice is to present confidence intervals for all models (these are presented in the main text but not in the appendix). This would make the data much easier to include in a meta-analysis, etc..., increasing the long-term impact of the analysis.

Reviewers' Comments & Our Responses

Reviewer #2 (Remarks to the Author):

The manuscript is much improved in response to reviewer comments. I am mostly satisfied with the responses and revisions. A few questions remain, which I am happy to leave to the judgment of the authors and editor:

- I still don't understand why the authors did not include humans and livestock as separate variables rather than combining into a single "human activity" variable. It seems potentially policy-relevant to assess whether the impact on downstream diarrheal disease is related to humans or livestock or both.

Diarrheal disease is caused by a host of bacterial, viral and parasitic organisms, most of which are spread by faeces-contaminated water. Water contaminated with human faeces is of particular concern, but animal faeces also contain microorganisms that can cause diarrhea. In our focus on effects of forests, we wanted to have the single strongest measure of all such contamination (and typically people and livestock co-occur). Thus, we aggregate these two sources into a single variable to jointly control for upstream conditions that are sources of pathogens and microorganisms that cause diarrheal disease. This helps make a single, strong comparison to areas that are lower human presence, and allows us to do a single sample split into areas of high and low human pressure.

- It would be useful to show in a supplemental table the correlation between variables used in the model to assure that there is not a problem of co-linearity

Thank you for this helpful suggestion. We have incorporated the table in the supplementary material (Table SI.11) and discuss it in lines 534-539 in the main text. We found low to moderate correlation coefficients (absolute values lower than 0.4) which should not be a major concern for our analysis.

- I still think the title would be more appropriate if it said "childhood diarrheal disease" rather than the broad term "children's health". The paper describes an important result that deserves publication.

Because diarrheal disease is associated to malnutrition and also a major cause of childhood mortality we think it reasonable to use the term childrens' health. Thus, we would prefer to keep "health" in the title.

Reviewer #3 (Remarks to the Author):

The authors have revised the manuscript and improved it in many ways. I do have one remaining concern that was raised in my initial review and not fully addressed.

In the main model (results shown in Fig 2) age, human activity and tree cover appear to have been modeled as continuous variables. I suggested they authors conduct a sensitivity analysis to evaluate the appropriateness of this modeling decision. In their revision, the authors have done this, but only for a subset of the data: Figure 5 presents estimates for age and upstream tree cover as categorical variables, but only for rural populations divided into populations with high vs. low upstream human activity. This is an interesting figure and allows exploration of interactions between upstream human activity and tree cover. However, as the marginal effects are based on the full model, the sensitivity analysis should be presented for the same data and model. Depending on the results of this analysis, the findings could be presented in the Supplementary Information at your discretion. But, in the absence of this information supporting the modeling of your exposure variables as continuous, I don't think the marginal effects estimates, which appear to be the main findings of the paper, are

justified.

Thank you for this constructive idea for examining the robustness of our results. We have included the sensitivity analysis for the full model in the supplementary material in Table SI.10 and point to it in the main text (527-531). This analysis supports the results of our main comparison between urban and rural households. Upstream tree cover is only significant for rural households and for levels above the 75th percentile.

Minor comment.

The table titles and notes in the Supplementary Appendix could be revised to describe the information presented more thoroughly. It would be nice, for example, to describe in a footnote what information is provided in the parentheses in tables S1.1 and S1.3. Good epidemiological practice is to present confidence intervals for all models (these are presented in the main text but not in the appendix). This would make the data much easier to include in a meta-analysis, etc..., increasing the long-term impact of the analysis.

Agreed, that is helpful perspective on our communications there. We now clarify that the numbers in parentheses are the standard errors of the coefficients, which can be used to calculate the confidence intervals that we agree are useful for the average reader and beyond. In terms of how to communicate that information, given that these tables already have a lot of information in them we believe we could keep the standard errors as they provide the necessary information to calculate confidence intervals that can be used for meta-analysis.

Reviewers' Comments:

Reviewer #3:

Remarks to the Author:

The authors have responded to the comments adequately.